# *SPOP* expression is associated with tumor-infiltrating lymphocytes in pancreatic cancer

Xiao Juan Yang[‡], Yong Feng Xu[‡], Qing Zhu[ORCID]*

Abdominal Oncology Ward, Cancer Center, West China Hospital of Sichuan University, Chengdu, Sichuan, P.R. China

‡ XJY and YFX should be considered joint first author.
* newzhuqing1972@yahoo.com

## Abstract

### Background

Speckle Type POZ Protein (SPOP), despite its tumor type-dependent role in tumorigenesis, primarily as a tumor suppressor gene is associated with a variety of different cancers. However, its function in pancreatic cancer remains uncertain.

### Methods

SPOP expression and the association between its expression and patient prognosis and immune function were evaluated using The Cancer Genome Atlas (TCGA), Genotype-Tissue Expression (GTEx), The Tumor Immune Estimation Resource 2.0 (TIMER2.0) database, cBioportal, and various bioinformatic databases. Enrichment analysis of SPOP and the association between SPOP expression with clinical stage and grade were analyzed using the R software package. Then immunohistochemistry (IHC) was used to estimate the correlation between SPOP and tumor-infiltrating lymphocytes (TILs) in patients with pancreatic cancer.

### Results

As part of our study, we assessed that SPOP was anomalously expressed in kinds of cancers, associated with clinical stage and outcomes. Meanwhile, SPOP also played a crucial role in the tumor microenvironment (TME). The expression level of SPOP was significantly correlated to tumor-infiltrating immune cells (TICs) in pancreatic cancer.

### Conclusions

Our study uncovered the potential corrections in SPOP with TICs, suggesting that SPOP may act as a biomarker for immunotherapy in pancreatic cancer.

**Data Availability Statement:** All relevant data are within the paper and its Supporting Information files.

**Funding:** The author(s) received no specific funding for this work.

**Competing interests:** The authors have declared that no competing interests exist.

# 1 Introduction

Cell neoplastic transformation is a multi-step process which includes the acquisition of some certain characteristics and these characteristics confers a survival advantage in their microenvironment [1]. These traits, coined as the hallmarks of cancer, are acquired as neoplastic cells evolve and enable them to be tumorigenic and malignant [2]. Tumor microenvironmental (TME) has been reported as a critical trigger to promote tumor progression, associated with aggression, metastasis, and recrudescence [3–5]. However, with the constant progress and development of public repositories in recent years, such as The Tumor Immune Estimation Resource 2.0 (TIMER2.0) dataset [6], it is possible to analysis pan-cancer genes and evaluate corrections in these genes with clinicopathologic prognosis and TME.

Speckle Type POZ Protein (SPOP), a well-known E3 ubiquitin ligase adaptor protein, plays a significant role in DNA damage repair, knockdown of SPOP spontaneous results in a series of replication stress and impaired recovery from replication fork stalling [2]. Although SPOP has been considered as a tumor suppressor in pancreatic cancer by targeting NANOG [7], however, the potential immunological role of SPOP in pancreatic cancer have rarely been estimated holistically. Hence, this manuscript is aimed at comprehensively estimating the expression profiles, immune cells, and genomic characteristics of SPOP in multiple cancers including pancreatic cancer and assessing the association of SPOP with infiltrating immune cells (TICs) in pancreatic cancer.

# 2 Methods

## 2.1 RNA-sequencing data and bioinformatic analysis

The overall expression levels of Speckle Type POZ Protein (SPOP) in cancers, we utilized The Tumor Immune Estimation Resource 2.0 (TIMER2.0, http://timer.cistrome.org) [6], and Gene Expression Profiling Interactive Analysis (GEPIA, http://gepia.cancer-pku.cn/) to estimate the expression profiles and immune scores of SPOP in various kinds of cancer [8, 9]. TIMER 2.0, is a sophisticated repository that integrated multi-gene expression profiles and generalized functions of tumor-infiltrating immune cells (TICs) of various patients in 23 kinds of cancer from The Cancer Genome Atlas Program (TCGA) [10]. GEPIA, another exhaustive online analysis database, provided a robust algorithm for 23 kinds of tumors, has been utilized to estimate pan-cancer survival and clinicopathological data based on UCSC (http://xena.ucsc.edu/cite-us) [11]. Then in order to analyze the survival and other clinicopathological data among cancers, those normalized RNA-seq data and relevant prognosis materials for 23 types of cancers were acquired from TCGA. In those data, patients with incomplete clinical data were excluded when the specific clinical factors were not available. Furthermore, the overall survival curves of 23 kinds of tumors were explored by the survival packages by R software. The Kaplan-Meier analyzing method was applied to design relative survival curves. The lower and higher 50% of gene expression levels were chosen as the analytical standards. The statistical significance was measured by using Log-Rank with a *P*-value significant threshold of 0.05.

## 2.2 Immunohistochemistry (IHC) staining

To estimate the different expressions of SPOP at the protein profiles in patients' tissues. Immunohistochemistry (IHC) images of SPOP protein levels in 32 paired human pancreatic adenocarcinoma (PAAD) tissues and normal tissues were observed. Besides, 72 human PAAD tumor tissues were used to estimate the correlation between SPOP and CD4+ T cells as well as CD8+ T cells. The tissue samples were obtained from the pathology platform of West China Hospital and had been reviewed by the ethics department (CAT.202203V4). Specimen were

embedded with paraffin, the thickness of tissue samples was sectioned into 3mm, then those sections were deparaffinized in xylene and rehydrated through different concentrations of alcohols (100%, 80%, and 50%). 3% hydrogen peroxide was used to inhibit the activity of endogenous peroxidase. Several monoclonal antibodies (CD4 and CD8 come from protein-tech, Cat. No 67786-1-Ig, and 66868-1-Ig) were used as the reaction at a dilution of 1:250 over-night at 4°C. After washing the sections in PBS, we incubated them with the biotinylated secondary antibody for 30 min at 37°C. Diaminobenzidine (DAB) solution was used to color sections as brown staining. For negative control, 1% BSA/PBS was used in place of the primary antibody, and was processed in the same manner. Cells with <10% staining were scored as negative staining (-, 0); cells with 10–49% staining were scored as (+, 2); cells with 50–74% staining were scored as (++, 3); and cells with 75–100% staining were scored as (+++, 4). The staining color was scored as negative staining particle (-,0); light-yellow particle (+, 1), brown-yellow particle (++, 2), and brown particle (+++, 3). Finally, we estimate final score by the area of positive cells staining number score added the staining color score [12].

## 2.3 SPOP-related gene enrichment analysis

The potential molecular mechanism of the SPOP and co-expressed genes in tumorigenesis were measured by Pearson's correlation index. According to the expression scores, those genes which |log2fold-change| > 0.5 and -log10 (p-value) > 1 were deemed as significant genes, and then showed the results via volcano plots. Next, those significant genes have also been analyzed in the functional enrichment study of Gene Ontology (GO) gene sets and Kyoto Encyclopedia of Genes and Genomes (KEGG) by the R software (version 4.05), those potential biological activities and signaling pathways were discovered by the GGPLOT packages.

Furthermore, to estimate the potential genes of protein-protein interaction in SPOP, the relative co-expression genes of SPOP were also filtered out by another database, Search Tool for the Retrieval of Interacting Genes (STRING, https://cn.string-db.org/) is an influential plat-form, served for calculating the functional protein to protein interactional networks [13]. All Protein-Protein Interaction (PPI) pairs with an interactive score of >0.4 were adopted. High-score nodes play a crucial role in ensuring the reliability of the PPI network. The degree of all complex networks was visualized by Cytoscape (v3.9.1, https://cytoscape.org/) [14].

Gene set enrichment analysis (GSEA) is a sophisticated analytical method for evaluating gene expression profiles and determining the statistical significance of signal pathways [15]. On the basis of the median level of SPOP expression, samples were divided into two phenotype subgroups, then, gene sets were estimated to identify common biological functions of SPOP. The significant gene sets met the following criterions: 1. p-value < 0.05; 2. | logFC | > 1.

## 2.4 Biological analysis and tumor microenvironment

The Tumor Immune Estimation Resource 2.0 (TIMER 2.0) platform, collecting of survival information from TCGA, comprehensively investigates the correlation between the abundance of TICs and types of cancer by ESTIMATE (Estimation of Stromal and Immune cells in Malig-nant Tumor tissues using Expression data). TIMER 2.0 using a sophisticated algorithm to investigate relationships between SPOP expression profiles and immune cells or relative mole-cules of pan-cancer. Analyzing the TICs of B cell, CD8+ T cell, CD4+ T cell, T cell regulator (Tregs), natural killer cell (NK cell), neutrophils, and mast cells. Besides, the overall survival outcomes correlation between cancer patients and myeloid dendritic cell as well as B cell were also evaluated.

The relationship between SPOP expression profiles and immunotherapy-related genes, ther-apeutic targets, and immune scores in the tumor microenvironment was investigated via an

excellent online database, called SangerBox (http://sangerbox.com). Data from SangerBox was evaluated to analyze the tumor mutational burden (TMB) or microsatellite instability (MSI) in pan-cancer microenvironment, and calculated by GGPLOT packages in R software.

## 2.5 Repository used to discover SPOP genomic mutation in pan-cancer

cBio Cancer Genomics Portal (c-BioPortal) (http://cbioportal.org), an automatic deduction website for analysis of all-around cancer genomic characters datasets, was applied to estimate multidimensional cancer genomic mutational materials [16]. The correlation between SPOP genomic mutation frequency and related survival data exploration were also examined by Kaplan-Meier survival analysis.

## 2.6 Statistical analysis

R 4.0.5 (https://www.R-project.org/), GraphPad Prism 5 and SPSS Statistics 26 were used for statistical analysis. Vitro data were expressed as mean ± SEM, and Students' t-test was applied for calculating P values. As for statistics from clinicopathologic parameters and IHC, the chi-square test and Fisher's exact test were applied to analyze the association between SPOP expression and TILs. *indicates P< 0.05, **indicates P< 0.01, ***indicates P< 0.001.

## 2.7 Statements

The time that the data were accessed for research purposes is 20th March, 2023. And we can not access to information that could identify individual participants during or after data collection.

## 2.8 Ethics statement

The study was conducted in compliance with the International Conference on Harmonization guidelines for Good Clinical Practice (E6) and the 2013 Declaration of Helsinki. The study was approved by the institutional review board of Sichuan University and the consent from parents or guardians was waived by the ethics committee

## 3 Results

### 3.1 Transcriptional levels of SPOP in pan-cancer patients

Tumors are characterized by heterogeneity, with a variety of substances presenting diverse expressions among them, therefore, we first verified the overall expression levels of SPOP in different forms of malignancies, after analyzed the transcriptional profiles of SPOP in 23 different types of cancer by using a free online database called TIMER2.0 (Fig 1A), SPOP was found to exhibit divergent expression levels in numerous cancers. The SPOP transcriptional expression profiles among Cholangiocarcinoma (CHOL) and Liver hepatocellular carcinoma (LIHC) were revealed to be significantly downregulated compared with their para-cancer tissues. In some cases, para-cancer tissue mRNA information was insufficient in the TCGA database, so we further estimated the overall expression differences of SPOP between cancer and para-cancer tissues combined with the TCGA and GTEx databases. These findings clarified that Adrenocortical carcinoma (ACC), Bladder Urothelial Carcinoma (BLCA), Cervical squamous cell carcinoma and endocervical adenocarcinoma (CESC), Colon adenocarcinoma (COAD), Esophageal carcinoma (ESCA), Lung adenocarcinoma (LUAD), Lung squamous cell carcinoma (LUSC), Ovarian serous cystadenocarcinoma (OV), Prostate adenocarcinoma (PRAD), Rectum adenocarcinoma (READ), Skin Cutaneous Melanoma (SKCM), Stomach adenocarcinoma (STAD), Testicular Germ Cell Tumors (TGCT), Thyroid carcinoma (THCA), Uterine Corpus Endometrial Carcinoma (UCEC), and Uterine Carcinosarcoma

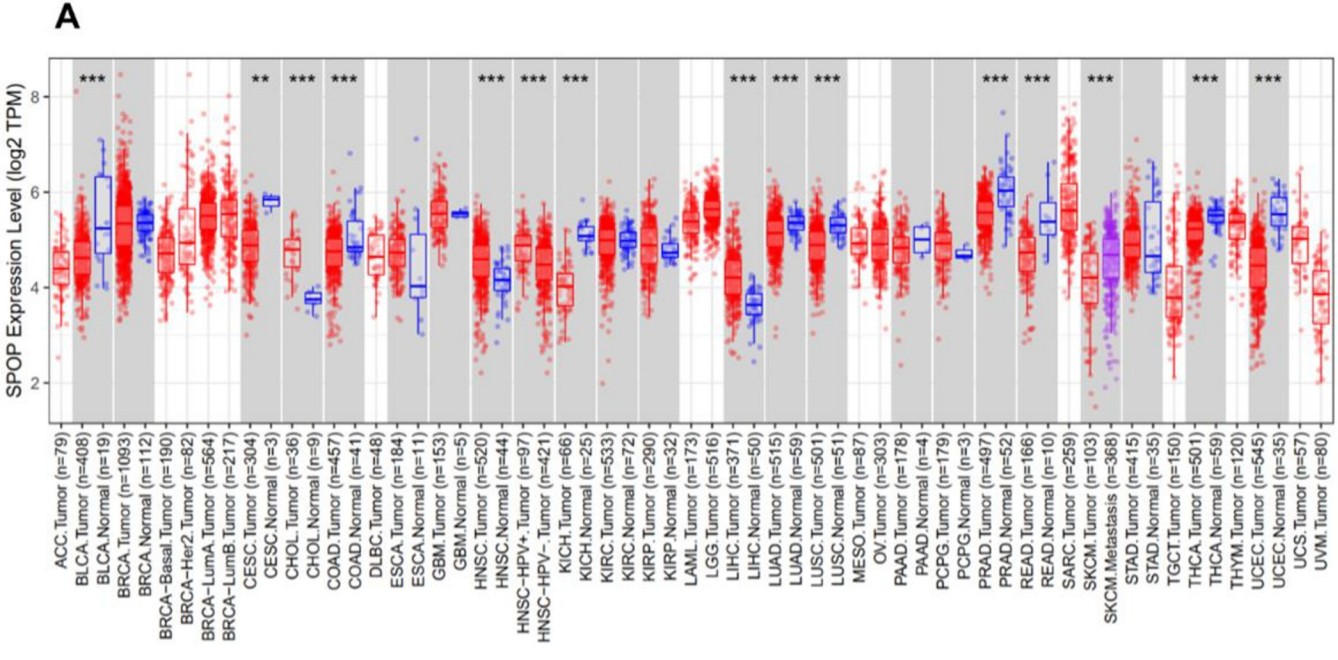

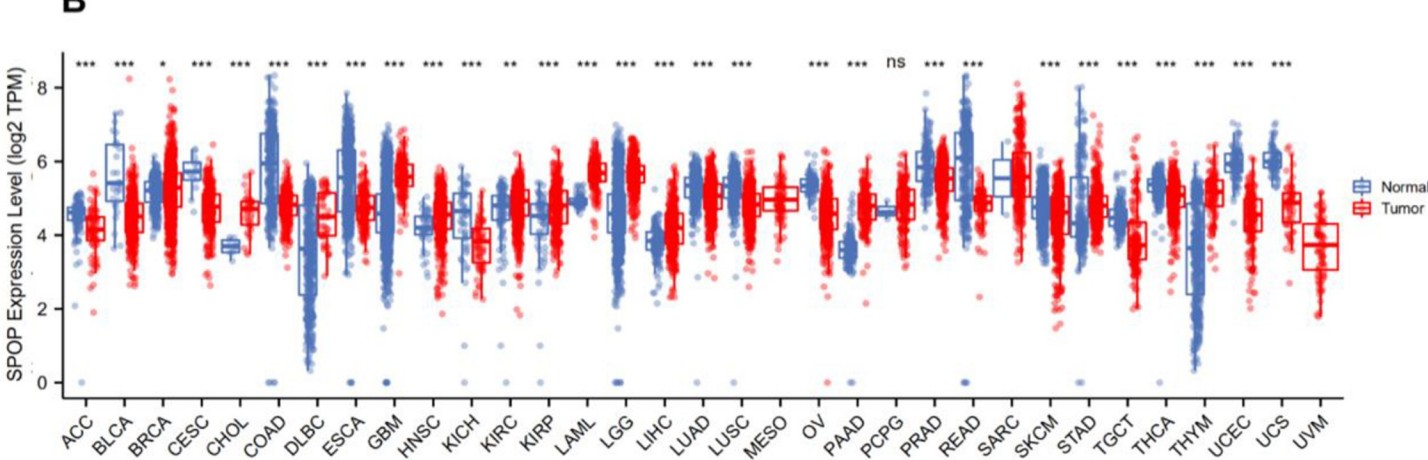

**Fig 1. Expression levels of Speckle Type POZ Protein (SPOP) in various cancers.** a, SPOP mRNA expression profiles in human 23 kinds of cancers from the TIMER 2.0 repository and (b) TCGA + GTEx databases. (*P < 0.05, **P < 0.01, ***P < 0.001).

(UCS) showed a low-level expression of SPOP compared with para-cancer tissues (Fig 1B). The result indicated that most of cancer group (16/23) had a significantly lower SPOP level than normal group.

## 3.2 The expression of SPOP is correlated with clinicopathological features of pan-cancer

The relationships between the relative SPOP mRNA expression profiles and different clinico-pathological characters of pan-cancer patients were downloaded from the TCGA dataset and performed in R software. The results revealed that SPOP mRNA expression profiles correlated significantly with AJCC/UICC TNM (Tumor Node Metastasis) stage in patients with BRCA,

COAD, ESCA, LUAD, Head and Neck squamous cell carcinoma (HNSC) and STAD (Fig 2A–2F). Meanwhile, the associations between SPOP mRNA expression and histologic grade were calculated and consequences presented that SPOP mRNA expression level was correlated to grades in GBMLGG, OV, Pancreatic adenocarcinoma (PAAD) (Fig 2G–2I). In addition, we tested the receiver operating characteristic (ROC) curves to evaluate how well SPOP as a biomarker is capable of discriminating between individuals in pan-cancer. The area under curves (AUC) were performed in types of cancer (S1 Fig). Those data indicated that lower SPOP expression profiles were significantly associated with advanced stage or grade in multi-type cancers.

## 3.3 The expression of SPOP is correlated with prognosis in pan-cancer patients

Based on the average expression levels of *SPOP*, patients were separated into high-expression or low-expression subgroups. The prognostic value of *SPOP* in human cancers was observed in TCGA repository, we found that a higher *SPOP* mRNA expression profiles were related to favorable overall survival (OS) in CESC (HR: 0.62, 0.39–1.00, P = 0.05, Fig 3A); LUAD (HR: 0.68, 0.50–0.91, P = 0.01, Fig 3B); OV (HR: 0.68, 0.50–0.93, P = 0.014, Fig 3C); PAAD (HR: 0.58, 0.36–0.95, P = 0.03, Fig 3D); Sarcoma (SARC) (HR: 0.56, 0.34–0.92, P = 0.021; Fig 3E) and SKCM (HR: 0.52, 0.38–0.69, P < 0.001; Fig 3F), and also had higher progression-free survival (PFS) in BRCA (HR: 0.57, 0.41–0.80, P = 0.001, Fig 3G); COAD (HR: 0.69, 0.49–0.98, P = 0.036, Fig 3H) and Colon adenocarcinoma and Rectum adenocarcinoma (COADREAD) (HR: 0.70, 0.51–0.94 P = 0.02, Fig 3I);. The above phenomenon proved that the expression of *SPOP* may play an outstanding biomarker to identify the prognosis profiles in pan-cancer.

## 3.4 Genomic characteristics

The highly frequency of somatic gene mutations are responsible for the progression of human cancers, leading to a wild range of functional and biological changes [17], such as tumorigenesis and metastasis [18, 19]. Since SPOP plays a significant role in tumor infiltration, the correlation between genetic stability in human tumor samples and SPOP should also be taken into account. Tumor mutational burden (TMB) and microsatellite instability (MSI) have been reported as an excellent biomarker to evaluate therapeutic efficiency [20, 21], and prognosis of tumor patients [22], thus, the correlation between TMB, MSI, and SPOP was estimated, our consequences presented that SPOP expression significant positive correlated with MSI in Acute Myeloid Leukemia-like (LAML) and TGCT, significant negative correlations within BRCA, DLBC, HNSC, KICH, KIPAN, LGG, OV, PRAD, STAD, STES, and THCA (Fig 4A). For TMB, significant negative relations in BRCA, CHOL, KIRP, LIHC, STAD, and STES (Fig 4B). Meanwhile, SPOP genomic alterations in multi carcinoma were discovered by the cBio-Portal database. The outcomes presented that the top 5 mutational frequency of SPOP mutation arise in endometrial cancer, breast carcinoma, embryonal cancer, pancreatic cancer, and prostate cancer (Fig 4C). To further validate the role of SPOP in mutation-related survival, based on the mutation of SPOP, patients were divided into altered (n = 21) and unaltered (n = 261) groups, a poorer survival outcome has been observed in the altered group (Fig 4D). Next, we further concentrate on genes co-mutated with SPOP, a surprising result has been observed that nearly all genes have a positive co-mutated correlation with SPOP (Fig 4E), in other words, when SPOP has been altered, most of genes also be mutated. In conclusion, SPOP has been described as a significant tumor-suppression role in multicancer, our bioinformatic analysis revealed that mutated SPOP leaded to a poor survival, suggesting that tumors

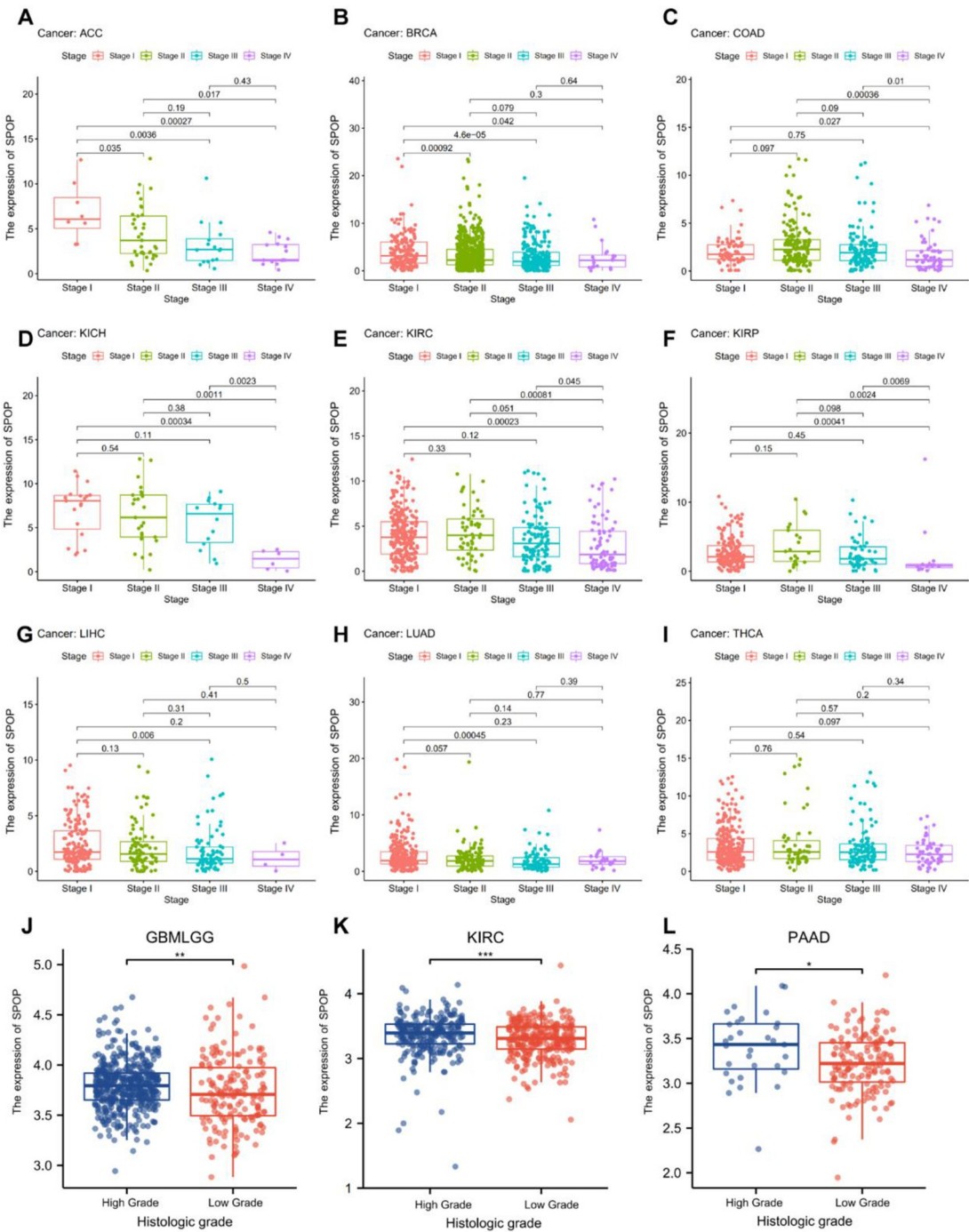

**Fig 2. Pan-cancer SPOP clinicopathologic expression analysis.** a–f, The SPOP mRNA expression levels between different clinicopathological TNM stages in Breast invasive carcinoma (BRCA), Colon adenocarcinoma (COAD), Esophageal carcinoma (ESCA), Lung adenocarcinoma (LUAD), Head and Neck squamous cell carcinoma (HNSC) and Stomach adenocarcinoma (STAD) from TCGA database. g-i, The SPOP mRNA expression levels between early and advanced histological grades in Glioma (GBMLGG), Ovarian serous cystadenocarcinoma (OV) and Pancreatic adenocarcinoma (PAAD). (*P < 0.05, **P < 0.01, ***P < 0.001).

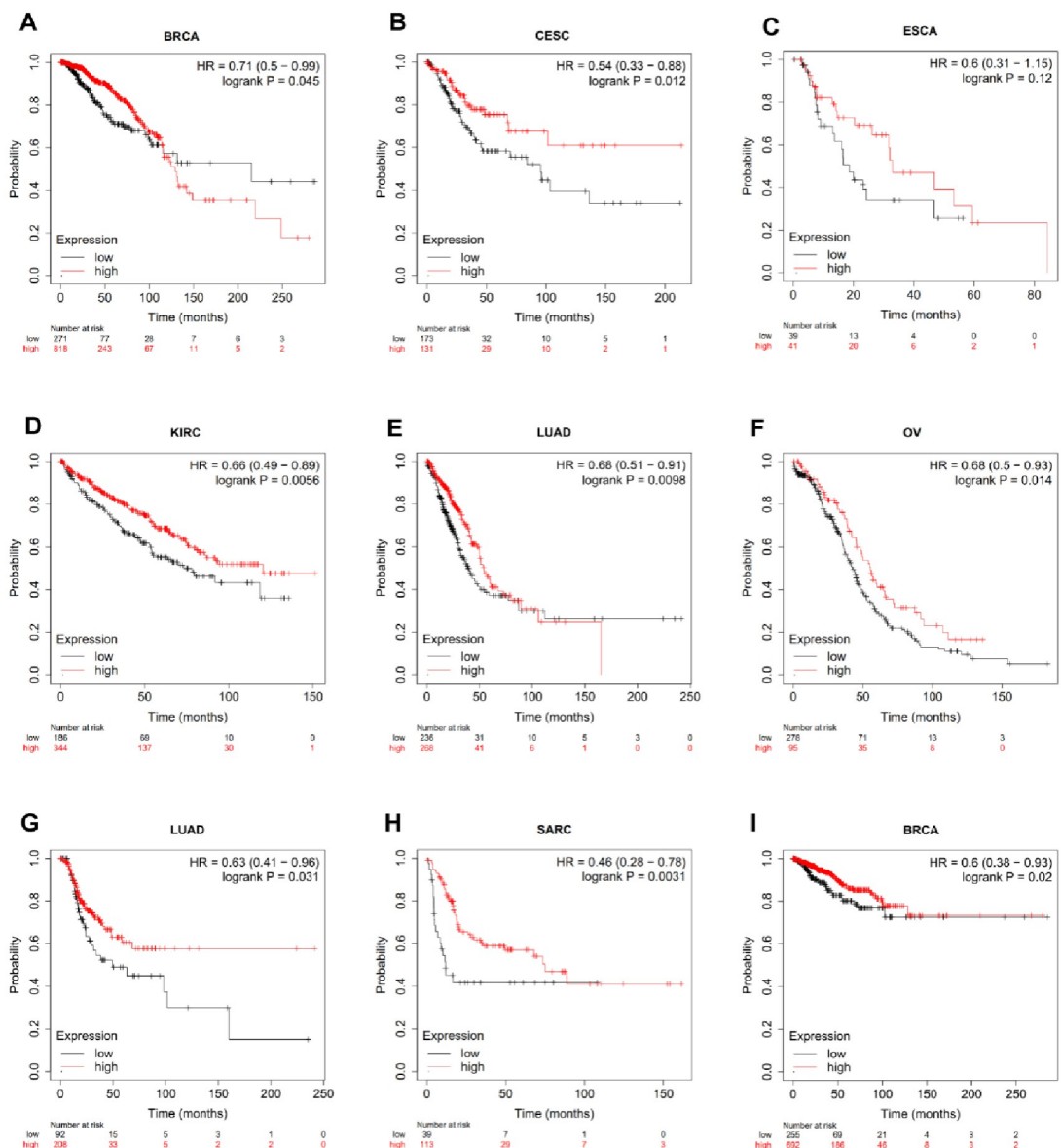

**Fig 3. The association between SPOP expression and cancer patient prognosis.** a-f, Kaplan-Meier survival curve analysis of the correlation between SPOP expression and overall survival (OS) in Breast invasive carcinoma (BRCA), Cervical squamous cell carcinoma and endocervical adenocarcinoma (CESC), Liver hepatocellular carcinoma (LIHC), Lung adenocarcinoma (LUAD), Ovarian serous cystadenocarcinoma (OV) and Pancreatic adenocarcinoma (PAAD). g-i, The progression-free survival (PFS) in Breast invasive carcinoma (BRCA), Colon adenocarcinoma (COAD) and Colon adenocarcinoma and Rectum adenocarcinoma (COADREAD).

try their efforts to immune evasion strategy and recruit SPOP to provide a protective microenvironment against complement-mediated lysis.

## 3.5 SPOP-related gene enrichment analysis in pancreatic cancer

SPOP-related gene enrichment materials were collected from the TCGA platform to further evaluate biological functions of SPOP and SPOP-correlated genes for relevant pathways. GO and KEGG enrichment studies were carried out in R software using a program called cluster

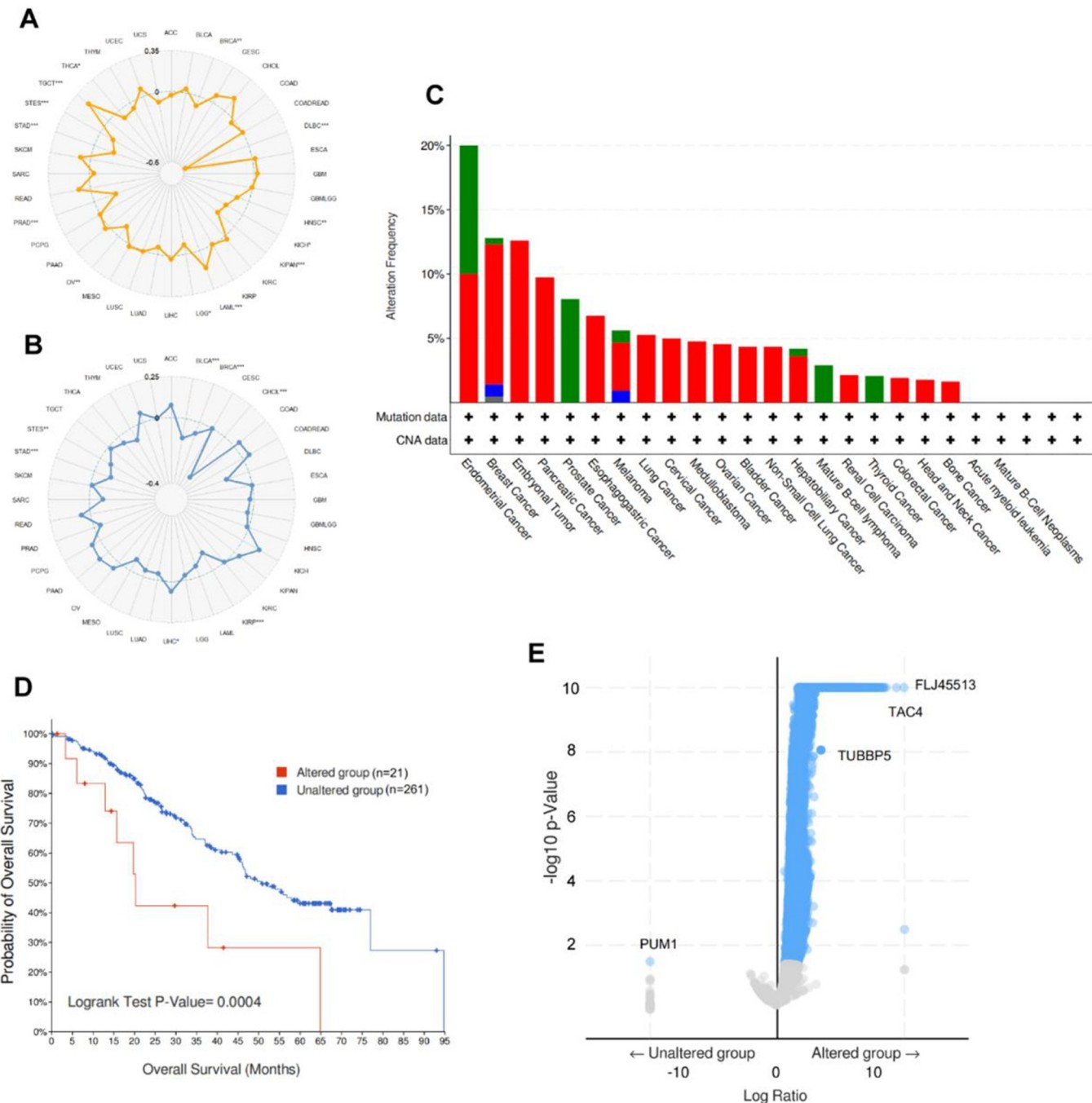

**Fig 4. The genomic characteristics of *SPOP*.** Relationship between *SPOP* expression with microsatellite instability (MSI) (a) and tumor mutational burden (TMB) (b) were exhibited by radar plot. c, Holistic view of mutation subtypes in multi cancer. d, survival outcome of altered (n = 28) or unaltered (n = 254) *SPOP* group. e, co-mutated genes with *SPOP*.

Profiler, enrich plot, and ggplot2. Only results with p-values, q-values no more than 0.05, and |logFC| > 1 were deemed as significantly enriched genes. Next, we gathered the top 100 most significant genes and analyzed the potential gene enrichment function (Fig 5A and 5B). Interestingly, the GO examination presented that those significant genes were enriched in a series of immune-related functions, such as "T cell activation", "neutrophil degranulation", "immune

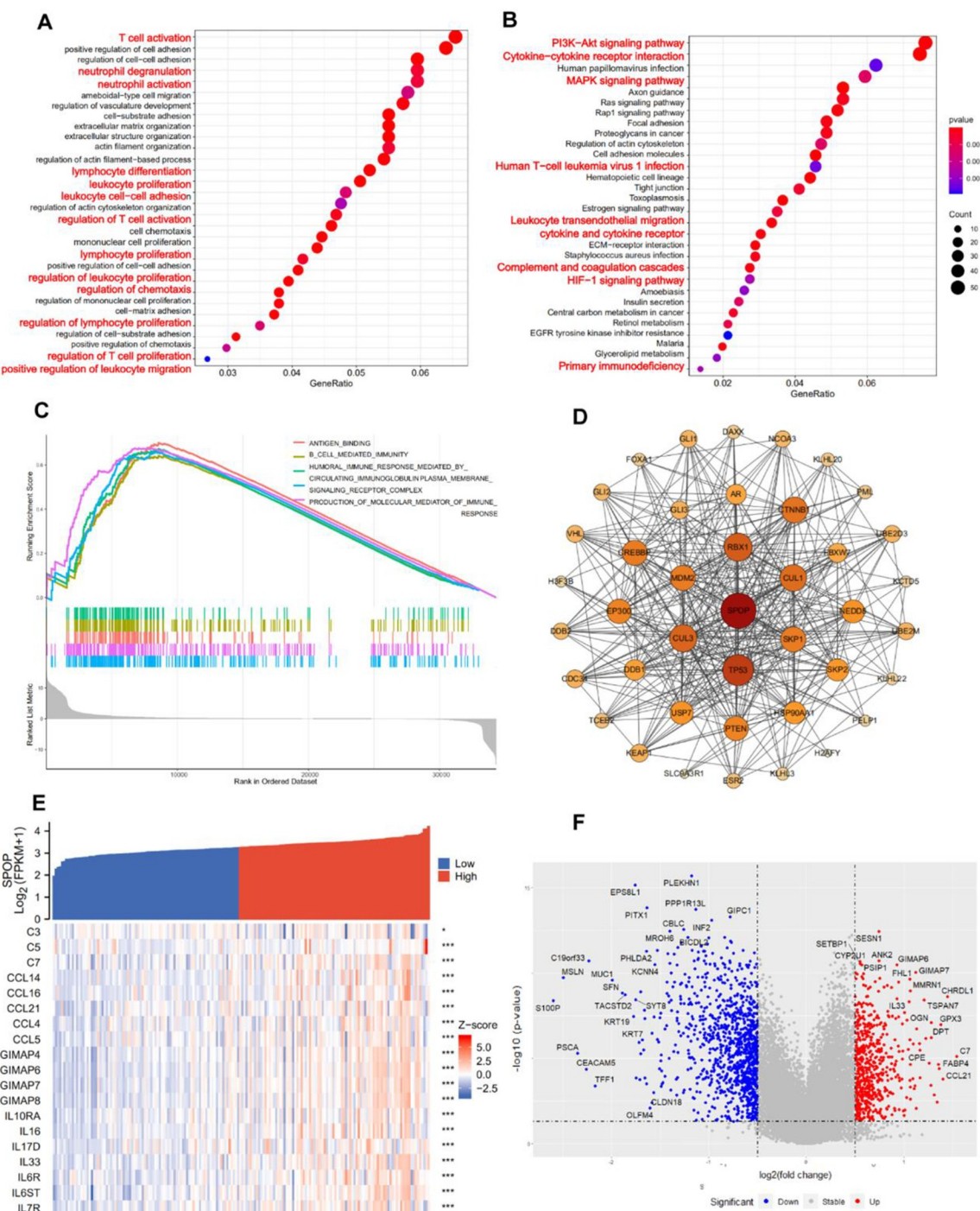

**Fig 5. Gene enrichment analysis of SPOP-related hub genes.** a, Significant Gene Ontology (GO) signal pathways of the top 100 most significant genes with SPOP. b, Significant Kyoto Encyclopedia of Genes and Genomes (KEGG) most positively related to SPOP. c, Gene set enrichment analysis (GSEA) functional annotation of SPOP. Curves marked with varied flags mean various signal pathways in cancer. d, the network of top 50 associated genes with SPOP were obtained from STRING website. e, Heatmap for the correlation of SPOP and immune-related genes. f, Volcano plot exhibit significantly differentially expressed genes, positively-related genes marked by red, and negatively-related genes marked by blue. (*P < 0.05, **P < 0.01, ***P < 0.001).

response", "lymphocyte differentiation", and "leukocyte proliferation". Meanwhile, The KEGG results also indicated that "PI3K−Akt signaling pathway", "MAPK pathway" and "Cytokine−cytokine receptor interaction" may associate with the carcinogenic mechanism of SPOP. The PI3K-AKT and MAPK pathway have previously been reported to be related to immune escape [23, 24], suggested that the potential role of SPOP in immune function. To further explore the underlying mechanism, we constructed Gene Set Enrichment Analysis (GSEA) based on the expression of SPOP, as expected, SPOP enriched in a series of important immune-related signaling pathways, such as "antigen binding", "B cell mediated immunity", "humoral immune response mediated by circulating immunoglobulin", and "production of molecular mediator of immune response" (Fig 5C). In addition, we examined the top 50 significant SPOP-binding proteins interactions and obtained those experimental data from the STRING network (Fig 5D). Furthermore, the Volcano plot exhibits significantly differentially co-express genes was shown. According to the significant gene result, the plot showed that there were 1641 differentially expressed genes (DEGs) which contained 602 upregulated DEGs (colored by red) and 1039 downregulated DEGs (colored by blue) (Fig 5E). Interestingly, a series of immune function-related genes, play significant roles in complement activation and T cell immune regulation, upregulated in the volcano plot (colored by red), such as IL33, CCL21, GIMAP6, GIMAP7 etc (Fig 5F). The above findings suggest that SPOP might be a useful biomarker of immunological function in pancreatic cancer.

## 3.6 Correlation between SPOP expression profile and tumor-infiltrating immune cells in pancreatic cancer

Since, the functional gene enrichment analysis suggested that SPOP involved in immune regulation in pancreatic cancer. We next to evaluate the correlation of SPOP levels with tumor-infiltrating immune cells (TICs). CIBERSORT, provided a sophisticated procedure to evaluate 23 types of TICs, was investigated to further confirm the correlation between SPOP expression and the immune component, when the SPOP expression profile was split into high and low groups, the immune cells showed a high correlation with SPOP expression levels in PAAD (Fig 6A), including CD4+ T cell (R = 0.491), CD8+ T cell (R = 0.624), myeloid dendritic cell (R = 0.548), neutrophils (R = 0.451), T cell regulator (Tregs) (R = 0.568), macrophage (R = 0.499), natural killer cell (NK cell) (R = 0.470), and mast cells (R = 0.419) (Fig 6B–6I). Finally, we evaluated SPOP expression level in the component of TICs by a single cell database called TISCH. The result indicates that SPOP have a co-expression in B cell, CD8+ T cell, and other types of immune cells (Fig 6J and 6K).

Furtherly, correlation analysis between SPOP and TICs were conducted in PAAD. We evaluated the immune cell infiltration scores in PAAD patients from TCGA database and exhibit by chord diagram (**Fig 7A**). The infiltration scores of T cell, B cell, and DC cell were significantly enrich in high-expression SPOP group (**Fig 7B–7E**). Finally, we estimated the impact of TICs and SPOP on overall survival outcome of cancer patients by TIMER2.0. The result indicates that high expression of SPOP profiles with higher CD4/8+ T infiltrating cells were correlated with a better survival in cancer patients (**Fig 7F and 7G**). Besides, Considering the significant immune character of SPOP in the T cell activation, and lymphocyte differentiation, we hypothesized that altered SPOP expression levels may impact the TICs expression profile. Thus, The TIMER, CIBERSORT, XCELL, QUANTISEQ, and EPIC algorithms were used to systematically investigate the relationship between TICs level and SPOP expression in 23 different types of cancer. Interestingly, we revealed a positive infiltration value between SPOP expression and the estimated of B cell, CD4+ T cell, as well as CD8+ T cell in the PAAD (**Fig 8A–8C**).

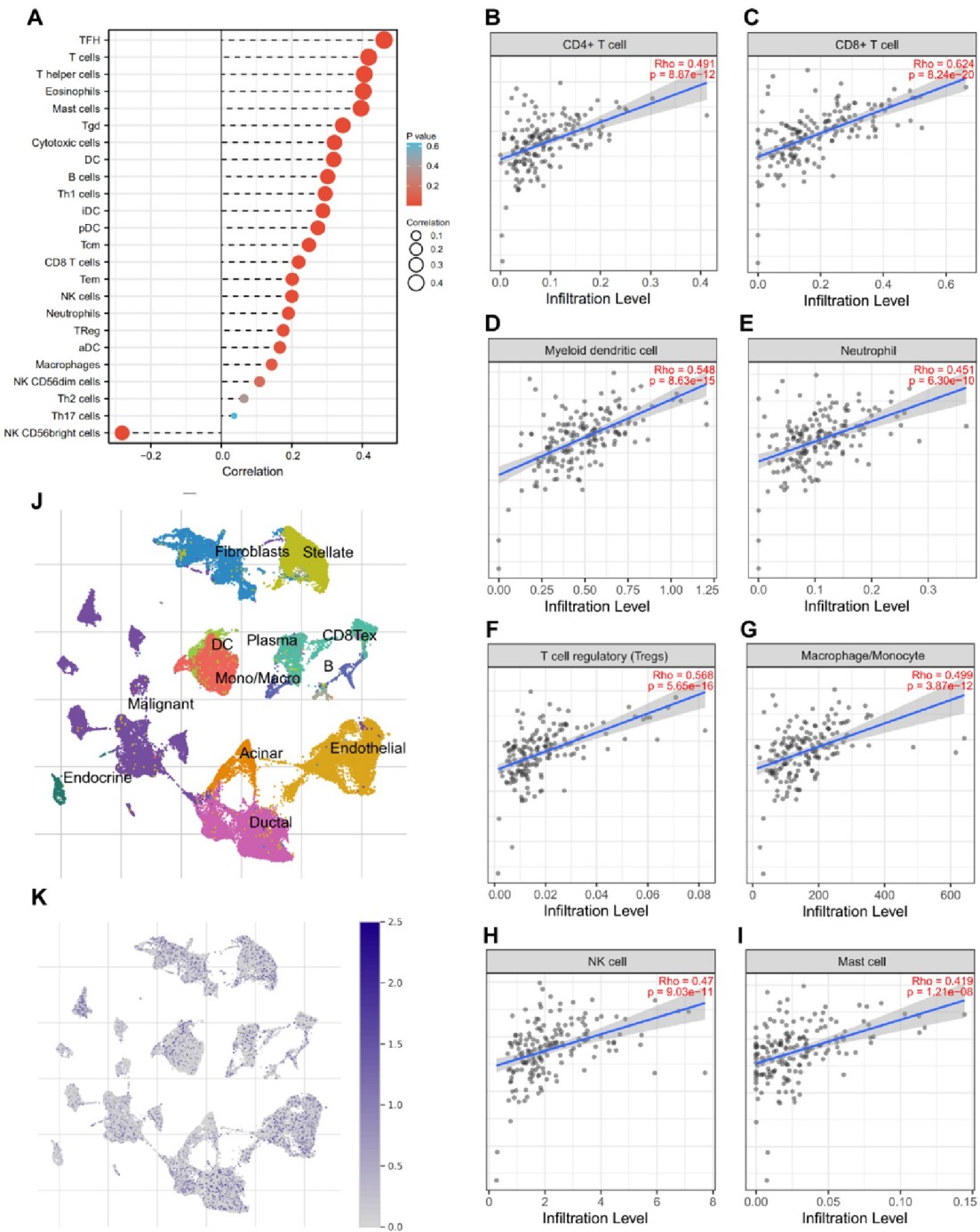

**Fig 6. The correlations between SPOP expression profiles and tumor-infiltrating immune cells (TICs) in cancer.** a, 24 types of TICs were evaluated by the ESTIMATE platform. b-i, Correlation between SPOP expression and CD4+ T cell, CD8+T cell, myeloid dendritic cell, neutrophils, Tregs, macrophage, NK cell, and Mast cells. j, Overall view of single cell clusters. k, SPOP expression level in types of cells.

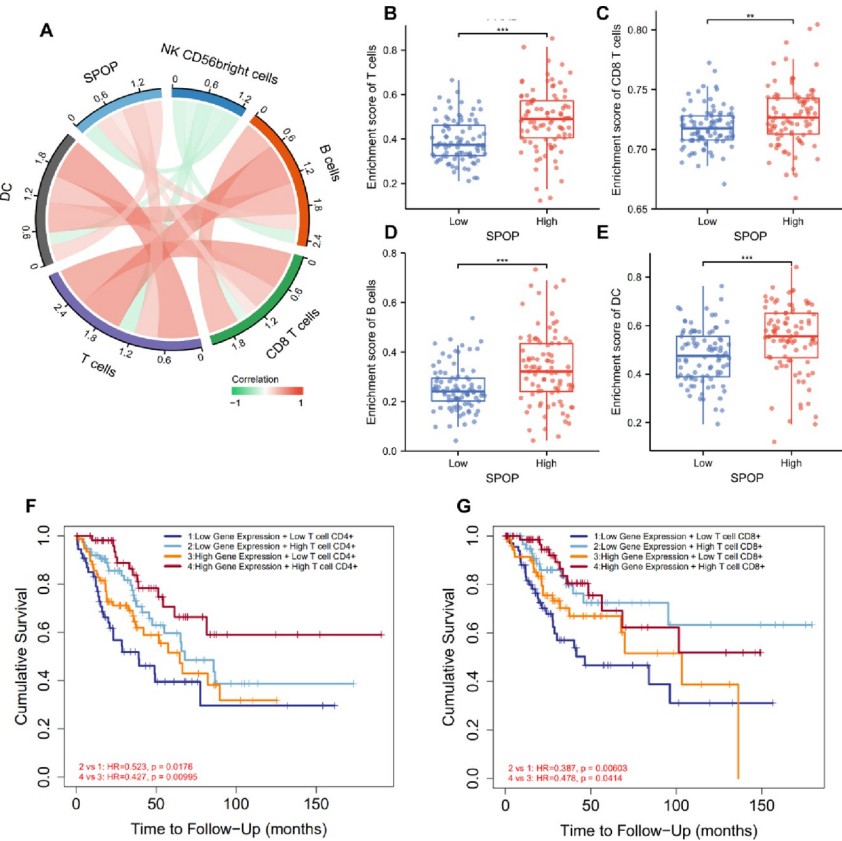

**Fig 7. The correlational analyses between infiltrating immune cells and SPOP.** a, the correlation between SPOP and the infiltrating immune cells score (red: positive correlation, green: negative correlation). b-e, The enrichment score of immune cells in high and low SPOP expression cohort. F, Clinical prognosis outcome of high-CD4+ T cell vs low-CD4 + T cell groups and (g) high-CD8+ T cell vs low-CD8+ T cell groups in cancer patients. *P < 0.05, **P < 0.01, ***P < 0.001.

## 3.7 Relationship between SPOP expression and tumor-infiltrating immune cells in patient tissues

Through the above bioinformatic enrichment analysis, we revealed that SPOP was mainly related to CD4+ T cell, CD8+ T cell, and neutrophils. We hypothesized that there might be some potential relationships between SPOP and tumor-infiltrating immune cells (TICs). Thus, we further evaluated whether the SPOP expression level was associated with TICs. We carried out immunohistochemistry staining in 72 human PAAD tumor tissues to estimate correlation between SPOP and CD4+ T cell as well as CD8+ T cell. As expected, SPOP staining scores were positively correlated with infiltration levels of CD4+ T cell (Fig 9A) ($R^2 = 0.5590$, $P < 0.001$), CD8+ T cell (Fig 9B) ($R^2 = 0.6526$, $P < 0.001$). The positive correlations were observed between SPOP expression and infiltration levels of CD4+ T cells and CD8+ T cells, implying the key role of SPOP in regulating tumor microenvironment.

## 3.8 The protein expression profile of SPOP in pancreatic cancer and normal tissues

Considering the complexity of post-transcriptional regulation, we further evaluate the protein expression level of SPOP by immunohistochemistry (IHC). In our cohort of 32 paired patients,

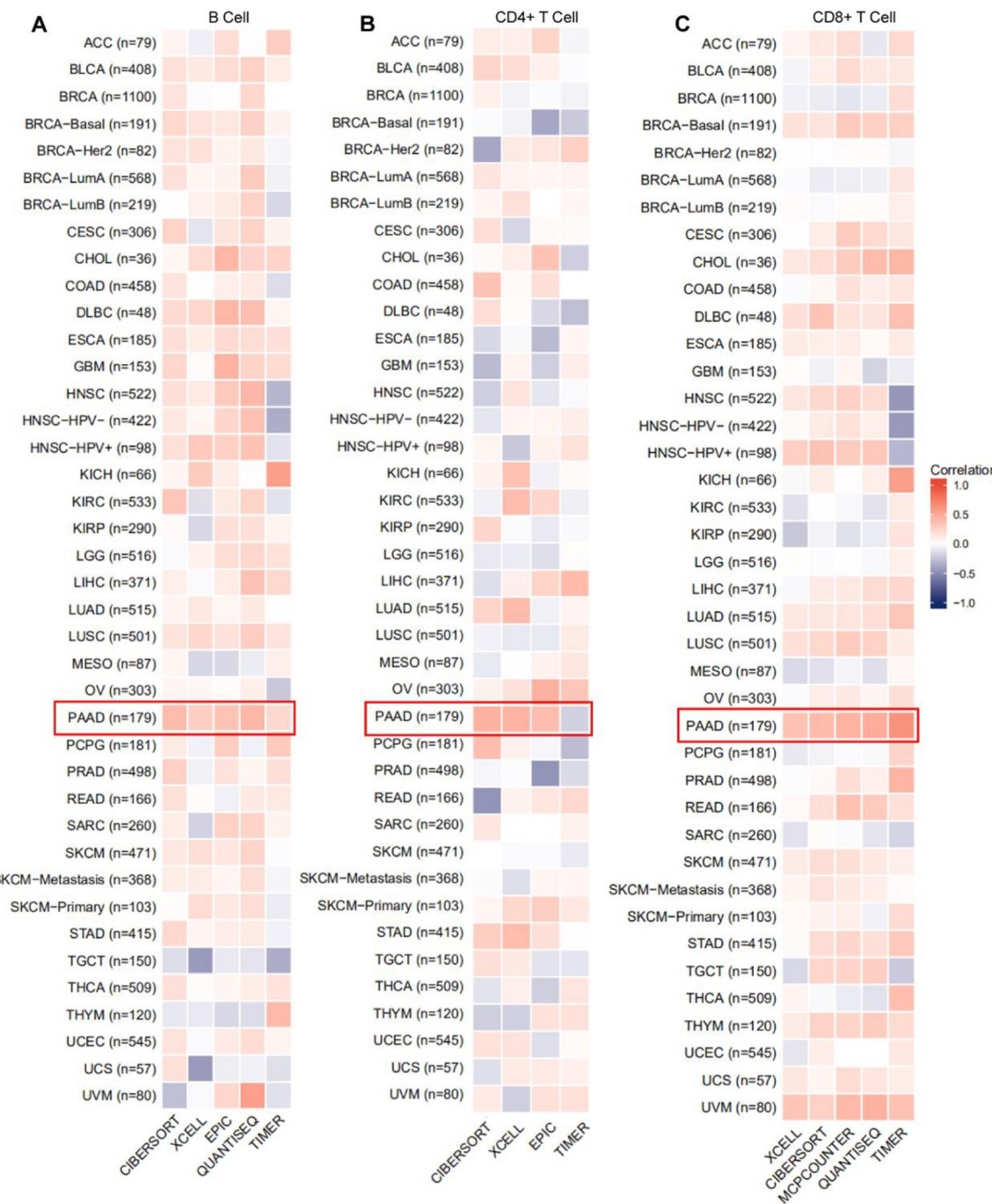

**Fig 8. The correlation between SPOP expression profiles and infiltration of tumor-infiltrating immune cells (TICs).** a, The correlative analysis of the SPOP expression level of TICs of B cell, (b) CD4+ T cell, and (c) CD8+ T cell in 23 types of cancer in TCGA (red: positive correlation, black: negative correlation).

the protein expression level of SPOP was downregulated in pancreatic cancer (PAAD) compared with normal tissues (P < 0.0001) (Fig 10A and 10B). Remarkably, its expression level was negatively correlated with poor histologic grade ($P = 0.0016$) (Fig 10C). In other words, the expression of SPOP has been observed a lower expression level in cancer tissues compared

with normal tissues, besides a lower expression of SPOP significantly correlated with advanced grade of cancer.

## 4 Discussion

The pan-cancer analysis gives a complete insight into the molecular abnormalities found in diverse malignancies, and also helpful for finding biomarkers for early detection and cancer-related target therapy [25]. Speckle Type POZ Protein (SPOP), a well-known E3 ubiquitin ligase adaptor protein, plays a significant role in DNA damage repair. Physiologically, several animal model experiments have been investigated the biologic function of SPOP in carcinogenesis. Just like Boysen and Claiborn et al reported, knockout of SPOP ($SPOP^{-/-}$) may present a congenital damage occur in brain, eye, and body formation of zebrafish, but this phenotype can be rescued by importing of human SPOP mRNA sequence artificially [26], besides, homozygous loss of $SPOP^{-/-}$ mice also found a postnatal lethality [27].

Moreover, there are still several reports recovered the significant role of SPOP in cancers. Many studies have been reported SPOP play a significant role in driving tumorigenesis in a variety of cancers, including prostate, breast, endometrium, liver, and colon [28–32]. Significantly, knockdown of SPOP leads to spontaneous replication stress and impaired recovery from replication fork stalling through PI3K/mTOR signaling mechanism to promote tumor progression [32]. Furthermore, as reported, a lower SPOP expression profile might serve as a potential biomarker for poorer prognosis in patients with colorectal cancer according to Kaplan-Meier survival curve analysis. On the contrary, if overexpressed SPOP in vitro, colorectal cancer cells can be dramatically repressed the proliferation and migration via EMT relative pathways, while this process can be reversed by knockdown of SPOP [33]. Additionally, a same result can be found in non-small cell lung cancer (NSCLC), a lower expression level of SPOP has been detected in NSCLC tissues compared with para-cancer tissues at both the transcriptional and translational levels, a decrease expression of SPOP was also considered a predictor of poor overall survival in patients with NSCLC, suggesting that SPOP could be a potential tumor suppressor in NSCLC [34]. In a word, downregulation of SPOP has been observed in cancer tissues compared with normal tissues, besides high expressed SPOP significantly correlated with a better prognosis of cancer, it's may due to an uncovered inhibiting mechanism to promote tumor growth.

In this present study, we firstly comprehensive verified that SPOP was downregulated expressed at transcriptional levels in ACC, BLCA, CESC, COAD, ESCA, KICH, LUAD, LUSC, OV, PRAD, READ, SKCM, STAD, TGCT, THCA, UCEC, and UCS cancers compared with para-cancer tissues, while SPOP was upregulated in DLBC, GBM, KIRC, LIHC, PAAD, and THYM, demonstrating that SPOP served as a potential biomarker in various cancers. Then we further explored the expression of the SPOP mRNA expression profiles, a negative correlation with TNM stage and grade in patients with ACC, BRCA, COAD, KICH, KIRC, KIRP, LIHC, LUAD, and THCA the results showed that a lower expression of SPOP always occurred in the advanced stage cancer. Meanwhile, a lower expression of SPOP generally predicts a poorer prognosis by Kaplan-Meier survival curve analysis in BRCA, CESC, ESCA, KIRC, LUAD, and OV. These results support SPOP as an outstanding prognostic biomarker of predicting tumor prognosis.

Human cancers progress due to a series of accumulation of genetic alterations [35, 36]. Thus, we further explored the SPOP genetic mutation in multiple cancers, the outcomes presented that the top 5 frequency of SPOP mutation existed in endometrial cancer, breast carcinoma, embryonal cancer, pancreatic cancer, and prostate cancer, those patients remained a poorer survival in mutated SPOP group. In addition, there is growing evidence suggests tumor

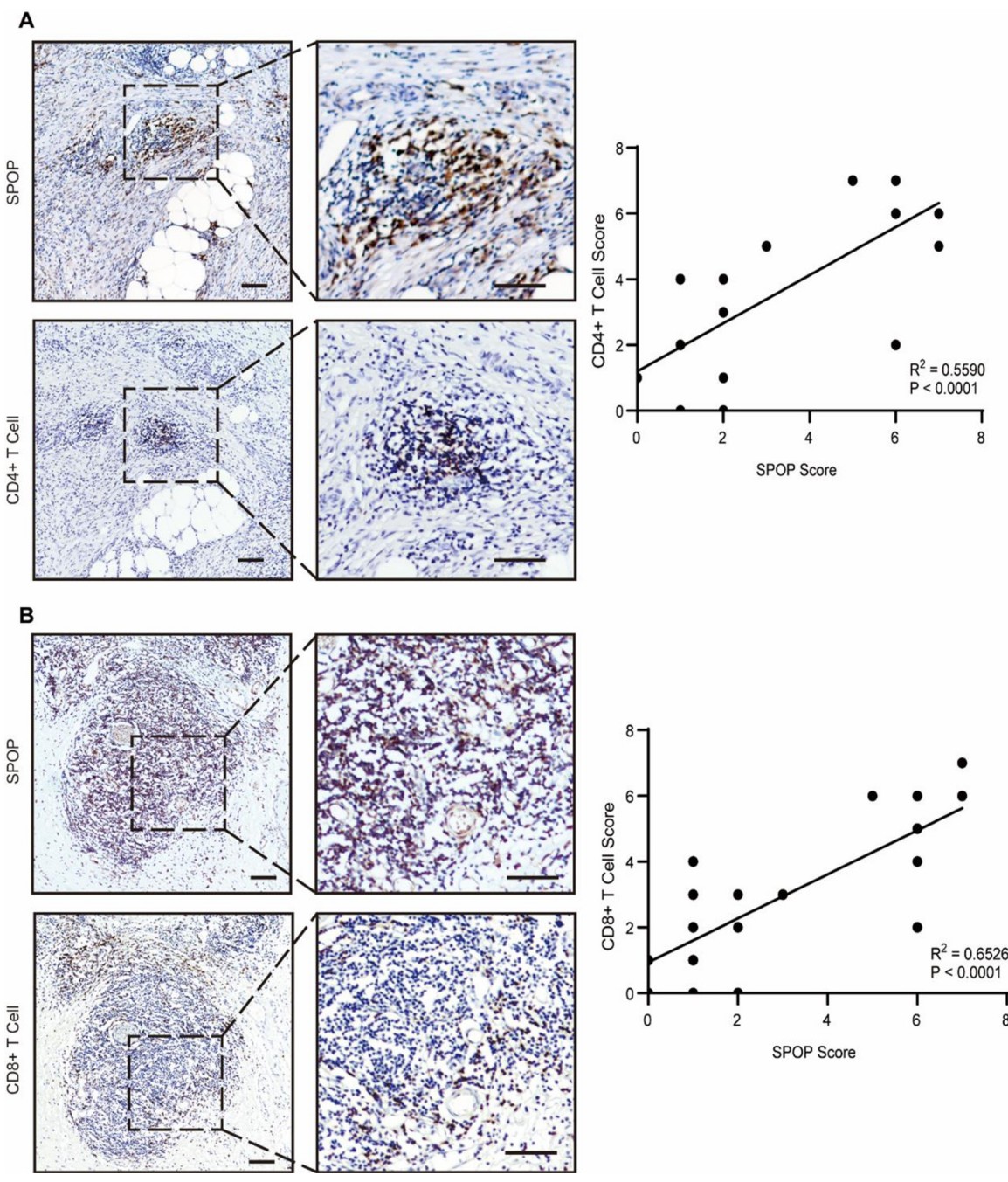

**Fig 9. TICs profile in PAAD samples and correlation analysis.** (a) The correlation expression level between SPOP and CD4 as well as (b) CD8 were tested in 72 paired PAAD patients. The Scatter plot showed the correlation of CD4 or CD8 proportion with the SPOP expression. Scale cars: 100 μm (left) and 20 μm (right).

microenvironment (TME) and tumor-infiltrating immune cells (TICs) have been proven to be an excellent effector in cancer patient prognosis and immunotherapeutic efficacy [37, 38]. In order to explore the potential function of SPOP involved in the cancerous progression, we further investigated the relationships between TICs and SPOP, the result indicated that almost all kinds of TICs (22/23, 95.7%) have a strong positive correlation with SPOP, such as CD4+ T

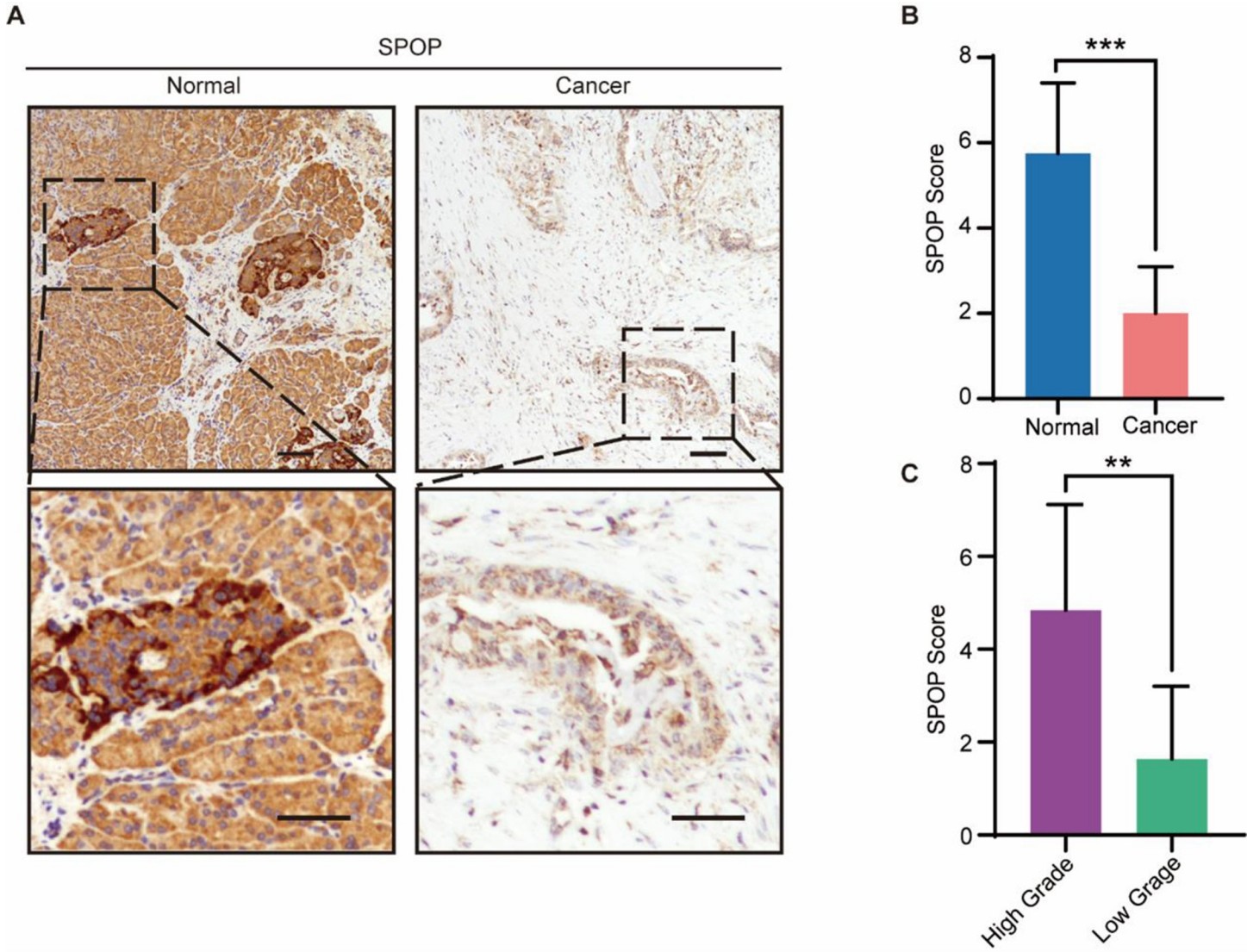

**Fig 10. SPOP expression in our PAAD samples.** (a) Comparison of SPOP gene expression between normal (left) and tumor tissues (right). (b) SPOP expressed in our PAAD and corresponding non-tumor pancreas tissues by IHC. (c) The analyze of patients' grade in PAAD tissues.

cell (R = 0.491), CD8+ T cell (R = 0.624), myeloid dendritic cell (R = 0.548), neutrophils (R = 0.451), T cell regulator (Tregs) (R = 0.568), macrophage (R = 0.499), natural killer cell (NK cell) (R = 0.470), and mast cells (R = 0.419). Furthermore, the GO and KEGG pathway analysis indicated that SPOP has a significant correlation with the immune system, such as "T cell activation", "neutrophil degranulation", "immune response", "lymphocyte differentiation", and "leukocyte proliferation". In a word, this result revealed that SPOP expression is significantly associated with immune function.

To further validate this immune correlation and elucidate the role of SPOP in cancer progression, we further assessed the correlation between SPOP expression level and CD4+ T cell as well as CD8+ T cell infiltration level by immunohistochemistry. Based on our analysis, SPOP staining scores were correlated with infiltration levels of CD4+ T cell ($R^2$ = 0.5590, P < 0.001) and CD8+ T cell ($R^2$ = 0.6526, P < 0.001). Besides, the expression level of SPOP and total protein of SPOP in the tumor tissues of PAAD is higher than the corresponding

control tissues. The positive correlations were detected between SPOP expression and infiltration levels of CD4+ T cells and CD8+ T cells, implying the key role of SPOP in regulating tumor immunology. Since, several studies indicated that a higher CD4+, CD8+ T cell infiltration scores were associated with better survival in cancers [39, 40], it is reasonable that high expression of SPOP may recruit more TICs and result in a better survival.

## 5 Conclusions

In conclusion, our study first provides a systematic investigation of SPOP in pan-cancer and revealed that SPOP expression levels decreased in types of cancer and related to overall survival. Besides, our results clarified a statistically positive correlation between SPOP expression and TICs in the pancreatic cancer, which implied that SPOP could be predicted the potential efficiency of immunotherapy, and served as a biomarker of pancreatic cancer.

## Supporting information

**S1 Fig. The AUC were performed in types of cancer.** (A-I) the ROC curve of Bladder Urothelial Carcinoma (BLCA), Cervical squamous cell carcinoma and endocervical adenocarcinoma (CESC), Acute Myeloid Leukemia-like (LAML), Lung squamous cell carcinoma (LUSC), Ovarian serous cystadenocarcinoma (OV), Pancreatic adenocarcinoma (PAAD), Thymoma (THYM), Uterine Corpus Endometrial Carcinoma (UCEC) and Uterine Carcinosarcoma (UCS).
(DOCX)

**S1 Dataset. Raw data.** Some of the results are analysed by R software as described in "Methods" section. This is the R programming language.
(ZIP)

## Acknowledgments

We thank all of our colleagues at the Sichuan University Cancer Center for constructive discussions and technical assistance. I thank XYF for drawing the images.

## Author Contributions

**Conceptualization:** Xiao Juan Yang.

**Data curation:** Xiao Juan Yang, Yong Feng Xu.

**Funding acquisition:** Qing Zhu.

**Investigation:** Xiao Juan Yang.

**Methodology:** Yong Feng Xu.

**Resources:** Yong Feng Xu.

**Software:** Yong Feng Xu.

**Visualization:** Qing Zhu.

**Writing – original draft:** Xiao Juan Yang.

**Writing – review & editing:** Xiao Juan Yang, Qing Zhu.

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
