## [Decision Letter · Decision Letter 0]

7 Feb 2024

PONE-D-23-40587SPOP Expression Is Associated with tumor-infiltrating lymphocytes in Pancreatic CancerPLOS ONE

Dear Dr. Zhu,

Thank you for submitting your manuscript to PLOS ONE. After careful consideration, we feel that it has merit but does not fully meet PLOS ONE’s publication criteria as it currently stands. Therefore, we invite you to submit a revised version of the manuscript that addresses the points raised during the review process. Based on the prossess review, the manuscript should grammarly improved and the points highlighted by reviewer 2 should be addressed.

We look forward to receiving your revised manuscript.

Kind regards,

Giovanni Messina

Academic Editor

PLOS ONE

Reviewers' comments:

Reviewer's Responses to Questions

**Comments to the Author**

1. Is the manuscript technically sound, and do the data support the conclusions?

Reviewer #1: Partly

Reviewer #2: Partly

2. Has the statistical analysis been performed appropriately and rigorously? 

Reviewer #1: Yes

Reviewer #2: No

3. Have the authors made all data underlying the findings in their manuscript fully available?

Reviewer #1: Yes

Reviewer #2: Yes

4. Is the manuscript presented in an intelligible fashion and written in standard English?

Reviewer #1: No

Reviewer #2: No

5. Review Comments to the Author

Reviewer #1: Authors present an interesting paper on the potential role of SPOP as a biomarker for immunotherapy in pancreatic cancer.

Manuscript is potentially interesting however, in its present form, it is not suitable for pubblication since english language needs extensive revisions. Please provide a more clear, grammarly correct and concise version of the manuscript.

Reviewer #2: 1. The author should claim an ethics statement with the supervision of ethics.

2. Figure 1A conflicts with Figure 1B regarding the cancer types that showed significant differences in the expression of SPOP mRNA levels. Timer 2 database stores the mRNA profiles from TCGA cohorts.

3. In some figure legends, there are different expressions regarding subplot nomination.

4. The author claimed SPOP as an outstanding prognostic

biomarker, but they did not test the robust prognostic value in different cohorts other than TCGA. Moreover, there were some much of cancer types.

5. In the figure legend of Figure 9, the author claims the TIC profile was obtained from 72 paired PAAD patients, while there were so few data points in the scatter plot.

6. The author should add scale information in the legend of Figure 10.

6. PLOS authors have the option to publish the peer review history of their article (what does this mean?). If published, this will include your full peer review and any attached files.

Reviewer #1: No

Reviewer #2: **Yes: **Yuchen Liu

---

## [Author Response · Author response to Decision Letter 0]

15 May 2024

Dear Editors and Reviewers:

Thank you for your letter and for the reviewers’ comments concerning our manuscript entitled “SPOP Expression Is Associated with tumor-infiltrating lymphocytes in Pancreatic Cancer (ID: PONE-D-23-40587)”. Those comments are all valuable and very helpful for revising and improving our paper, as well as the important guiding significance to our research. We have studied comments carefully and have made correction which we hope meet with approval. Revised portions are marked in red in the paper. The main corrections in the paper and the responds to the reviewer’s comments are as flowing:

Responds to the reviewer’s comments: 

Reviewer # 1

 Response to comment: Authors present an interesting paper on the potential role of SPOP as a biomarker for immunotherapy in pancreatic cancer. Manuscript is potentially interesting however, in its present form, it is not suitable for publication since English language needs extensive revisions. Please provide a more clear, grammarly correct and concise version of the manuscript.

 Response: Considering the Reviewer’s suggestion, we have read our paper carefully again. We have extensively revised English language including grammar through the whole article. We sincerely hope we have provided a more clear, grammarly correct and concise version of the manuscript. Thank you very much.

 Special thanks to you for your good comments.

Reviewer #2

1. Response to comment: The author should claim an ethics statement with the supervision of ethics.

 Response: It is really true as the Reviewer suggested that we should claim an ethics statement with the supervision of ethics, so we have added ethics statement to the Methods section of our manuscript according to the Reviewer’s comments. We have marked red in the Methods section.

2. Response to comment: Figure 1A conflicts with Figure 1B regarding the cancer types that showed significant differences in the expression of SPOP mRNA levels. Timer 2 database stores the mRNA profiles from TCGA cohorts.

 Response: Thank you for dedicating your time to review our manuscript and providing valuable feedback regarding its strengths and limitations. We sincerely apologize for any shortcomings identified within this article. Regarding the discrepancy observed in the expression levels depicted in Figure 1a and Figure 1b, we think that it stems from the utilization of data from the Timer 2.0 database, which primarily relies on TCGA database. It is worth noting that certain tumors or normal tissues within the sample set exhibit relatively small representation. For instance, pancreatic adenocarcinoma (PAAD) comprises only 4 cases of normal tissue. Recognizing the potential bias associated with such limited data, we have taken steps to enhance the robustness of our findings. Specifically, we have supplemented our dataset with additional normal tissue samples sourced from the GTEx database to ensure the reliability and validity of our analysis. We trust that these modifications have addressed the concerns raised and have strengthened the overall integrity of our study.

3. Response to comment: In some figure legends, there are different expressions regarding subplot nomination.

 Response: We are very grateful for your careful review and warm tips, and we unified the expressions in the article and Figure 4 Legend, which were all marked red according to your kind suggestions. 

4. Response to comment: The author claimed SPOP as an outstanding prognostic

biomarker, but they did not test the robust prognostic value in different cohorts other than TCGA. Moreover, there were some much of cancer types.

 Response: Thank you for sparing your time to read our manuscript and point out our limitations. We are so sorry for the shortcomings existed in this article. To improve the paper, on the one hand, we tested the prognostic value in the GSE cohorts and we have added it in the article and marked red. On the other hand, we have considered collecting various tumor tissue samples through a clinical multicenter collaboration to test SPOP expression levels and its prognostic value. We appreciate your suggestion, which will be a focal point in our forthcoming research endeavors.

5. Response to comment: In the figure legend of Figure 9, the author claims the TIC profile was obtained from 72 paired PAAD patients, while there were so few data points in the scatter plot.

 Response: Thank you for your warmhearted tips, we have carefully gone over the manuscript and rechecked our original data. We are very sorry for our mistake to write 72 paired PAAD patients. It should be 24 paired patients instead of 72 paired patients. Additionally, we have revised the description in the article and figure legend of Figure 9 to accurately reflect this clarification. We are very grateful for your constructive feedback on our manuscript.

6. Response to comment: The author should add scale information in the legend of Figure 10.

 Response: We are very sorry for our omissions about the scale information in the legend of Figure 10. We have added scale information according to the Reviewer’s comments. Added portion are marked in red in Figure 10. Thank you very much.

 Special thanks to you for your good comments.

We tried our best to improve the manuscript and made some changes in the manuscript. These changes will not influence the content and framework of the paper. And here we did not list the changes but marked in red in revised paper.

We appreciate for Editors/Reviewers’ warm work earnestly, and hope that the correction will meet with approval.

Once again, thank you very much for your comments and suggestions.

Sincerely，

Qing Zhu

---

## [Editor Report · Decision Letter 1]

27 Jun 2024

SPOP Expression Is Associated with tumor-infiltrating lymphocytes in Pancreatic Cancer

PONE-D-23-40587R1

Dear Dr. Qing Zhu,

We’re pleased to inform you that your manuscript has been judged scientifically suitable for publication and will be formally accepted for publication once it meets all outstanding technical requirements.

Kind regards,

Giovanni Messina

Academic Editor

PLOS ONE

Additional Editor Comments (optional):

I really appreciate the effort in improvement of the paper. The manuscript is now suitable for publication.
---

## [Editor Report · Acceptance letter]

17 Jul 2024

PONE-D-23-40587R1 

PLOS ONE

Dear Dr. Zhu, 

I'm pleased to inform you that your manuscript has been deemed suitable for publication in PLOS ONE. Congratulations! Your manuscript is now being handed over to our production team.

Kind regards, 

on behalf of

Dr. Giovanni Messina 

Academic Editor

PLOS ONE